# Impact of tumor location on oncological and perioperative outcomes after robot-assisted radical nephroureterectomy for upper tract urothelial carcinoma

Mahmoud Farzat[1,2]*, Ibrahim Altaie[2], Sami-Ramzi Leyh-Bannurah[3], Mykyta Kachanov[4], Florian M. Wagenlehner[1]

**1** Department of Urology, Pediatric Urology, and Andrology, Justus-Liebig University of Giessen, Giessen, Germany, **2** Department of Urology and Robotic Urology, Diakonie Klinikum Siegen, Siegen, Germany, **3** Martini-Klinik Prostate Cancer Center, University Medical Center Hamburg-Eppendorf, Hamburg, Germany, **4** Institute of Human Genetics, University Medical Center Hamburg-Eppendorf, Hamburg, Germany

* mahmoud.farzat@diakonie-sw.de

## Abstract

### Objective

to investigate the effect of tumor location on oncological outcomes in patients receiving robot-assisted radical nephroureterectomy (RANU) for upper urinary tract carcinoma (UTUC).

### Methods

A retrospective single-center cohort study of 54 patients with UTUC who underwent RANU by a single surgeon between July 2019 and July 2025, without neoadjuvant chemotherapy or previous or simultaneous cystectomy, were included. Patients were divided into two groups based on tumor location: 18 patients (33%) with ureteral tumors (Group 1) and 36 patients (67%) with renal pelvis tumors (Group 2). Demographics, perioperative data, and pathological results were analyzed. The primary endpoints Cancer-specific survival (CSS) and overall survival (OS) were estimated using Kaplan–Meier and univariable log-rank test.

### Results

Console time, blood transfusion, complications, and readmission were comparable in both groups. Group 1 experienced longer hospital stays (8 days vs. 6.5 days, p = 0.03). 48% of patients had ≥ pT2 disease, with a similar T-stage distribution across groups. Of 26 candidates for adjuvant therapy, 10 received chemotherapy with gemcitabine/cisplatin, 2 received nivolumab, and one patient received enfortumab vedotin with pembrolizumab. During a median 26.5-month follow-up, six patients developed

which permits unrestricted use, distribution, and reproduction in any medium, provided the original author and source are credited.

**Data availability statement:** The datasets generated and analyzed during the current study are not publicly available due to national regulations on the protection of sensitive personal information. Data can be requested from the Ethik-Kommission Westfalen-Lippe via email at gutachterkommission@aekwl.de. Data may be made available to eligible researchers upon completion of all required prerequisites, such as a Data Use agreement.

**Funding:** The author(s) received no specific funding for this work.

**Competing interests:** The authors have declared that no competing interests exist.

**Abbreviations:** URS, ureterorenoscopy; EAU, European Association of Urology; CDC, Clavien-Dindo Complication; UTUC, upper urinary tract carcinoma; RANU, robot-assisted radical nephroureterectomy; RARC, robot-assisted radical Cystectomy; LAD, Lymphadenectomy; LN, Lymph nodes; URS, ureterorenoscopy; AC, anticoagulation; NOAC, new oral anticoagulant; ASA, American Association of anesthesiology score; BMI, body mass index; Hgb, hemoglobin; PSM, positive surgical margins; CCS, cancer-specific survival; RFS, recurrence-free survival; OS, overall survival; BR, bladder recurrence; ICU, intensive care unit.

bladder recurrence, (median 9 months) after RANU (p = 0.10), and four developed distant metastases (median 4 months) (p = 0.72), resulting in a disease-free survival of 81% (p = 0.08)Cancer-specific survival was 94%, overall survival 89%, with no significant group differences (p = 0.24 and 0.49).

## Conclusion

In our series, we observed that tumor location does not impact postoperative and oncological outcomes after RANU for UTUC, regardless of adjuvant therapy. However, further studies are needed to explore this proposed hypothesis.

## Introduction

Regarding oncological outcomes, ureteral tumors demonstrate notably higher recurrence rates than tumors that originate in the renal pelvicalyceal system [1]. Krajewski et al. found in their meta-analysis involving 16836 patients that the location of UTUC in the ureter is associated with significantly poorer long-term oncological outcomes [2]. They even spoke in favor of neoadjuvant chemotherapy in ureteral UTUC patients [2]. On the other hand, Joshi and colleagues found that tumor size, rather than location, was associated with worse survival outcomes [3]. In their analysis involving 637 patients across multiple institutions, Yafi et al. found that the location of ureteral tumors, especially when combined with the multifocal disease in the renal pelvis, acts as an independent prognostic factor for increased recurrence of disease and cancer-specific mortality [4]. Other authors reported that the ureteral location is associated with a shorter metastasis-free survival [5]. Interestingly, Tanaka et al. found that ureter tumors, especially in the lower two-thirds, had a higher rate of local recurrence. In contrast, renal pelvic tumors had a higher prevalence of distant relapse in the lungs [6]. Additionally, multifocal presentation of ureteral tumors is a significant prognostic factor of the disease's progression-free survival [7]. Another study showed a link between tumor location and the occurrence of positive surgical margins [8]. Others suggest that tumor locations within the lower ureter are correlated with less favorable oncological outcomes [9]. Liu et al. reported a trend toward distal ureteral UTUC developing bladder recurrence, but not local recurrence or distant metastasis [10]. Moreover, two meta-analyses revealed a link between tumor location and intravesical recurrence [11,12]. However, Favaretto et al. found that tumor location did not predict bladder recurrence or cancer-specific survival [13]. Shibing et al. proved in their large multi-institutional cohort that tumor size over 3 cm correlated with worse recurrence-free, cancer-specific, and overall survival [14]. These findings were corroborated by subsequent case series [15]. Overall, most current research indicates that the location of ureteral tumors correlates with worse oncological results. This study examines the influence of tumor location on surgical and oncological outcomes in a contemporary, homogeneous cohort of patients undergoing robot-assisted nephroureterectomy (RANU) at a single center, performed by a single surgeon, and offers insights based on real-world clinical data.

## Methods

### Participants, exclusion criteria and study design

Between July 2019 and July 2025, 85 patients underwent UTUC surgery. In this retrospective context, we used non-probability sampling—specifically, convenience sampling. To achieve a more homogenous cohort, eight patients who underwent segmental ureterectomy, 15 who underwent simultaneous cystectomy, four who underwent prior cystectomies, two who underwent surgery in palliative settings, and two who underwent surgery after neoadjuvant chemotherapy for locally advanced cancer were excluded. This left 54 patients in our analysis, all undergoing surgery with curative intent (See Fig 1). Patients' data were assessed on the first of September 2025, to conduct a retrospective single-center cohort study. Based on tumor location, a total of 54 patients were divided into two groups: Group 1 included 18 patients with ureteral tumors, and Group 2 included 36 patients with renal pelvis tumors. Based on the final pathological results, patients with non-muscle-invasive UTUC (Ta, Tis, T1) were followed up according to EAU guidelines [16]. Patients with pT2 (tumor invades muscularis), or advanced disease pT3 (tumor beyond muscularis), or T4 (tumor invades adjacent organs or

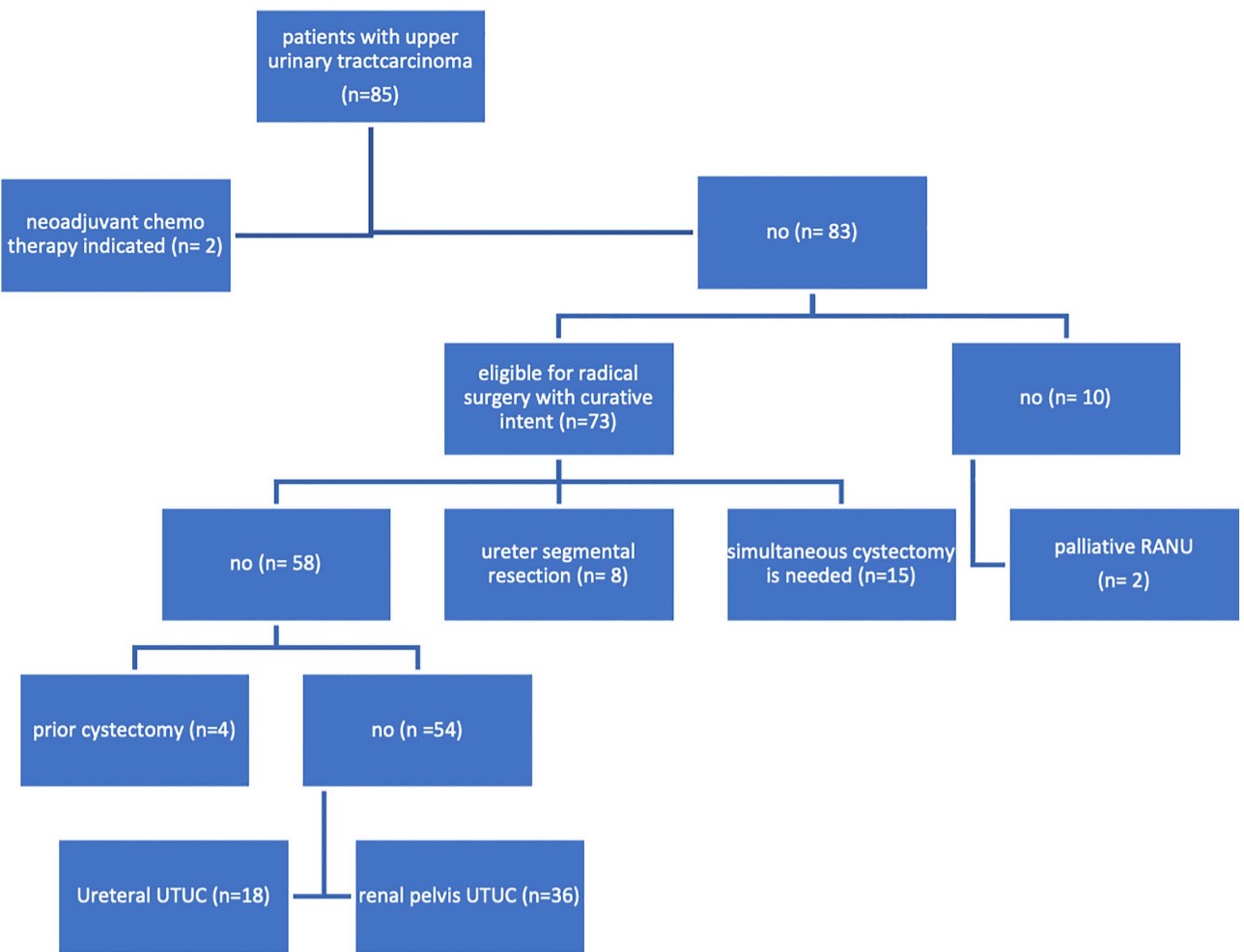

**Fig 1. Consort flow chart of the patients included in the study.**

through the kidney into perinephric fat), or N+ (lymph node involvement) were offered adjuvant platinum-based chemotherapy. Those who were ineligible for platinum therapy were given nivolumab.

## Preoperative diagnostic and surgical technique

Patients presenting with painless hematuria received a standardized diagnostic assessment, which included sonography, laboratory tests, and urethrocystoscopy. If bladder cancer was ruled out or found near the ureteral orifice, or if an upper urinary tract tumor was suspected, additionally, a retrograde pyelography and renal pelvic urine cytology were conducted. In those with tumor suspicion, a CT scan with a urinary phase was performed in all patients to facilitate better planning of the surgical intervention. Ureterorenoscopy (URS) with biopsy was performed in all patients when the diagnosis was undetermined via retrograde ureteropyelography and CT scan. Patients with locally advanced carcinoma who were eligible for adjuvant chemotherapy and willing to receive these treatments received their treatment before surgery.

All surgeries were performed by a single experienced robotic surgeon (>2,500 robotic procedures) using the Da Vinci X® system (Intuitive Surgical, Sunnyvale, CA, USA). All RANU patients were placed in a lateral position. An 8 mm/Hg Capnoperitoneum was created via a pararectal mini-laparotomy, which was later used to extract the specimen. Four robotic arms were employed. After completing the kidney procedure, the robotic instruments were redeployed for the ureteral part without requiring re-docking or repositioning of the patient. Patients with high-grade cytology or biopsy, evidence of local invasion on CT, variant histology, multifocal disease, tumor size ≥2 cm, or hydronephrosis were classified as high-risk UTUC and received Lymphadenectomy (LAD): ipsilateral iliac nodes in ureteral tumors and hilar, paraaortic, and interaortocaval nodes in renal pelvis tumors. All patients who consented received immediate intravesical instillation therapy with mitomycin 40 mg before transurethral catheter removal.

## Variables and statistical analysis

Clinical and oncological variables were prospectively collected in an institutional database and subsequently analyzed retrospectively after full anonymization. Demographic and perioperative data were analyzed, and postoperative complications within 90 days after surgery were classified according to the Clavien-Dindo classification [17]. Follow-ups were performed regularly according to EAU guidelines [16]. Data were analyzed with SPSS® v29. Categorical variables were reported as frequencies and percentages and compared using the Chi-square or Fisher's exact test, as appropriate. The Kolmogorov-Smirnov test was utilized to assess normal distribution. The independent T-test and Mann-Whitney U tests were applied to parametric and non-parametric variables. A p-value of < 0.05 was considered statistically significant. All p-values listed are two-tailed. Recurrence-free survival and CSS probabilities were estimated using Kaplan-Meier and univariable long-rank tests. The primary endpoint of the study was to assess how UTUC tumor location affects disease-free survival and overall survival. The research adhered to the ethical guidelines specified in the Declaration of Helsinki and received approval from the ethics committees of the Westfalen-Lippe Medical Association and the University of Muenster (2023–500-f-S). The need for patient consent is waived by the ethical committee since this is a retrospective analysis of existing patients' data. Nonetheless, all patients provided verbal informed consent for data from their medical records to be used in research for educational purposes.

## Results

### Baseline parameters

The median age of the patients was 73 years. Patients in the ureteral tumor group were more obese with a median BMI of 30.5 kg/m$^2$ versus a BMI of 27 kg/m$^2$ for the renal pelvis group (p = 0.04). 65% of patients were males. Patients presented with multiple comorbidities; however, ASA scores were comparable across groups. 39% of patients had cardiovascular disease, while Diabetes mellitus and COPD were less frequent. Anticoagulant use was more frequent in Group 1 (44%)

than in Group 2 (22%) (p = 0.31). Preoperative ureterorenoscopic definitive histological diagnosis was achieved in 55% of cases in the ureteral tumors group, compared with only 22% in the renal pelvis group (p = 0.01). While the overall hemoglobin values were normal in both groups, renal function showed relative deterioration, with a median value of 58 mL/min/1.73 m$^2$ (p = 0.3 and 0.4, respectively). Details are given in Table 1.

### Intra- and postoperative data

Console time was, on average, 10 minutes longer in Group 2. While Group 1 patients had a significantly longer mean hospital stay (8 days vs. 6.5 days), the median duration was 5 days across both groups (p 0 = .77). Overall, 52% of patients presented with non-muscle-invasive disease, 14.8% with pT2, and 33.3% with locally advanced tumors pT3-4, N +, exhibiting comparable distributions across the study groups (p = 0.39). High-grade tumors were observed more often in Group 2 (64%) (p = 0.17). Overall, n = 34/54 (63%) of patients in our cohort received immediate single-dose intravesical instillation of mitomycin, with a similar instillation rate across study groups (p = 0.33).

Out of 26 patients with muscle-invasive and locally advanced cancer, 13 received adjuvant therapy. Of them, 10 patients received three cycles of polychemotherapy gemcitabine/cisplatin. When renal function didn't allow a full dose, patients received a 60% or 80% reduction in cisplatin dose to improve tolerability, and in such cases, the dose was split over two days. 2 patients received Nivolumab for 1 year (4 weeks schema), and one patient was planned for adjuvant therapy with nivolumab, yet in the subsequent staging, new metastases were found and he received Enfortumab Vedotin in combination with Pembrolizumab.

**Table 1. Baseline characteristics and preoperative clinical and oncological parameters.**

| | Total (N = 54) | Ureter (N = 18) | Kidney Pelvis (N = 36) | p-Value |
|---|---|---|---|---|
| Age (years), median (IQR) | 73 (65 - 80) | 71 (63 - 78) | 74 (66 - 81) | 0.5 |
| Gender (Male), n (%) | 35 (64.8%) | 13 (72.2%) | 22 (61.1%) | 0.56 |
| BMI (kg/m²), median (IQR) | 28.0 (24.8 - 33.3) | 30.5 (26.6 - 37.8) | 27.0 (24.0 - 31.0) | 0.04 |
| ASA Score, n (%) | | | | 0.93 |
| 1 | 14 (25.9%) | 5 (27.8%) | 9 (25.0%) | |
| 2 | 14 (25.9%) | 4 (22.2%) | 10 (27.8%) | |
| 3 | 26 (48.1%) | 9 (50.0%) | 17 (47.2%) | |
| Preoperative Histology (URS), n (%) | | | | 0.01 |
| No Biopsy/ negative histology | 36 (66.7%) | 8 (44.4%) | 28 (77.8%) | |
| Ta/ G1 | 11 (20.4%) | 6 (33.3%) | 5 (13.9%) | |
| T1/ G2 | 4 (7.4%) | 3 (16.7%) | 1 (2.8%) | |
| T1/ G3 | 2 (3.7%) | 1 (5.6%) | 1 (2.8%) | |
| CIS (Tis) | 1 (1.9%) | 0 (0%) | 1 (2.8%) | |
| Cardiovascular Disease | 21 (38.9%) | 6 (33.3%) | 15 (41.7%) | 0.70 |
| Diabetes Mellitus II | 5 (9.3%) | 1 (5.6%) | 4 (11.1%) | 0.62 |
| COPD | 3 (5.6%) | 1 (5.6%) | 2 (5.6%) | 0.91 |
| Anticoagulation, n (%) | 16 (29.6%) | 8 (44.4%) | 8 (22.2%) | 0.31 |
| Aspirin (ASS) | 12 (22.2%) | 6 (33.3%) | 6 (16.7%) | |
| NOAC | 4 (7.4%) | 2 (11.1%) | 2 (5.6%) | |
| Preop. Hb (g/dL), median (IQR) | 13.1 (11.0 - 14.4) | 13.4 (12.2 - 14.4) | 12.9 (9.9 - 14.3) | 0.33 |
| Preop. GFR (ml/min), median (IQR) | 58 (42 - 70) | 62 (48 - 70) | 55 (41 - 70) | 0.48 |

Categorical data are presented as numbers %, BMI: body mass index, ASA: American association of Anesthesiologists, COPD: chronic obstructive pulmonary disease, NOAK: new oral anticoagulant, HB: hemoglobin, GFR: glomerular filtration rate.

All patients had negative surgical margins. LAD was performed in 35% of patients, with a median of 5 lymph nodes removed, and there was no significant group difference (p = 0.56). Only one patient in group 2 had lymph node metastasis. Four patients (7.4%) required perioperative transfusions. The overall median differences in hemoglobin and GFR of 2.1 g/dL and 9 mL/min/1.73$^2$ were similar across study groups (p = 0.82 and 0.91, respectively). Further details in Table 2.

## Complications, oncological and survival outcomes

The overall 90-day complication rate was 20% (n = 11/54), with no significant difference between the groups (27.8% vs. 16.7%, p = 0.4). Of the 11 complications, three were minor (Clavien-Dindo I-II) and eight were major (Clavien-Dindo III-V). The most common major complication was fascia dehiscence, which happened in 3 obese patients at the pararectal mini-laparotomy site, which was used to retrieve the specimen. One female patient with advanced peripheral artery disease developed an embolism in the iliac external artery on the same side of the RANU and underwent uneventful endovascular embolectomy. One male patient developed bleeding after surgery, necessitating a revision. Another patient with advanced COPD and a long history of constipation was reoperated on for bowel obstruction. We recorded two perioperative fatalities in our series. The first one was a female patient with an undiagnosed coagulation path disorder who died after a long stay in the intensive care unit (ICU). The other perioperative death involved an elderly male with extensive abdominal adhesions from prior colorectal cancer surgery. He underwent bowel resection at RANU start and died in the ICU after prolonged cardiac decompensation. More patients were readmitted in the ureteral tumor group (16.7%)

**Table 2. Intra- and postoperative results and pathological outcomes.**

| | Total (N = 54) | Ureter (N = 18) | Kidney Pelvis (N = 36) | p-Value |
|---|---|---|---|---|
| Console Time (min), median (IQR) | 70 (50 - 90) | 65 (41 - 85) | 75 (55 - 90) | 0.39 |
| Hospital Stay (days), mean (IQR), median | 7 (5 - 9) 5 | 8 (6 - 10) 5 | 6.5 (5 - 8) 5 | 0.77 |
| Pathological T Stage, n (%) | | | | 0.39 |
| Ta/T1/Tis | 28 (51.9%) | 11 (61.1%) | 17 (47.2%) | |
| T2 | 8 (14.8%) | 1 (5.6%) | 7 (19.4%) | |
| T3/T4 | 18 (33.3%) | 6 (33.3%) | 12 (33.3%) | |
| Tumor Grade, n (%) | | | | 0.17 |
| Low Grade (1/2) | 23 (42.6%) | 10 (55.6%) | 13 (36.1%) | |
| High Grade (3) | 31 (57.4%) | 8 (44.4%) | 23 (63.9%) | |
| Lymphadenectomy performed, n (%) | 19 (35.2%) | 6 (33.3%) | 13 (36.1%) | 0.90 |
| Lymph nodes removed, median (IQR) | 5 (2 - 10) | 7.5 (2 - 13) | 5 (2 - 9) | 0.56 |
| Lymph Node Metastasis, n (%) | 1 (1.9%) | 0 (0%) | 1 (2.8%) | 0.99 |
| Positive Surgical Margins | 0 (0%) | 0 (0%) | 0 (0%) | |
| Transfusion, n (%) | 4 (7.4%) | 1 (5.6%) | 3 (8.3%) | 0.50 |
| Intravesicale instillation therapy (single dose mitomycin 40 mg) | 34 (63%) | 13 (72%) | 21 (58%) | 0.33 |
| Adjuvant Therapy, n (%) | 13 (24.1%) | 4 (22.2%) | 9 (25.0%) | 0.99 |
| Gemcitabine/Cisplatin | *10 (18.5%)* | *3 (16.7%)* | *7 (19.4%)* | |
| Immunotherapy | *3 (5.6%)* | *1 (5.6%)* | *2 (5.6%)* | |
| Postop. Hb (g/dL), median (IQR) | 10.9 (9.5 - 12.4) | 11.2 (9.8 - 12.5) | 10.7 (9.2 - 12.3) | 0.40 |
| Δ Hb (g/dL), median (IQR) | −2.1 (−3.4 – −1.0) | −2.2 (−3.2 – −1.4) | −2.0 (−3.5 – −0.8) | 0.82 |
| Postop. GFR (ml/min), median (IQR) | 49 (37 - 59) | 52 (43 - 59) | 48 (34 - 59) | 0.31 |
| Δ GFR (ml/min), median (IQR) | −9 (−17 – −2) | −8 (−16 – −3) | −9 (−18 – −2) | 0.92 |

Categorical data are presented as numbers %, UTUC: Upper Urinary Tract Urothelial Cell Carcinoma, RARC: robot-assisted radical cystectomy. SD: standard deviation, CI: Checkpoint inhibitor, * Grade according to WHO classification 1999 (Busch et al.) [18].

compared to patients in the renal pelvis tumor group (8.3%). Over a median follow-up of 26.5 months, bladder recurrence was observed in 6 patients (11%), with a median time to recurrence of 9 months (p = 0.10). Four patients (7.4%) developed distant metastases (median time 4 months) (p = 0.72). This resulted in a disease-free survival of 81% (Fig 2), a cancer-specific survival of 94% (Fig 3), and, considering the 2 perioperative deaths, an overall survival of 89% (Fig 4). Statistically, both groups showed similar survival rates, p = 0.20, 0.24, and 0.49, respectively (Table 3).

## Discussion

Studies investigating tumor location, alongside other clinicopathologic factors influencing survival following RANU for UTUC, primarily focus on long-term outcomes and evaluate tumor location as a prognostic indicator of poorer survival [8,11,12]. However, the impact of tumor location on short-term surgical outcomes has not been thoroughly investigated.

The principal finding of our study is that tumor location (ureteral vs. renal pelvis) did not significantly affect short-term surgical or oncological outcomes following RANU for UTUC. Our observation aligns with findings from prior research [19]. In our cohort, 33% of patients were diagnosed with locally advanced (T3–T4, N+) disease, with exact incidence among study groups. Favaretto et al. (9) conducted a retrospective study involving 234 patients and found that pathological stage and nodal status were the only independent predictors of disease recurrence. Tumor location, on the other hand, did not affect outcomes. Similarly, our study found no significant associations between tumor stage, location, or nodal status and oncological outcomes. Our retrospective analysis included a homogeneous patient group, all of whom underwent the same surgical procedure, with similar rates of lymphadenectomy and adjuvant therapy. Under these conditions, tumor location did not appear to influence survival. However, understanding how chemotherapy or immunotherapy affects survival outcomes remains vital. In our study, 26 of 54 patients (48%) had muscle-invasive and locally advanced disease,

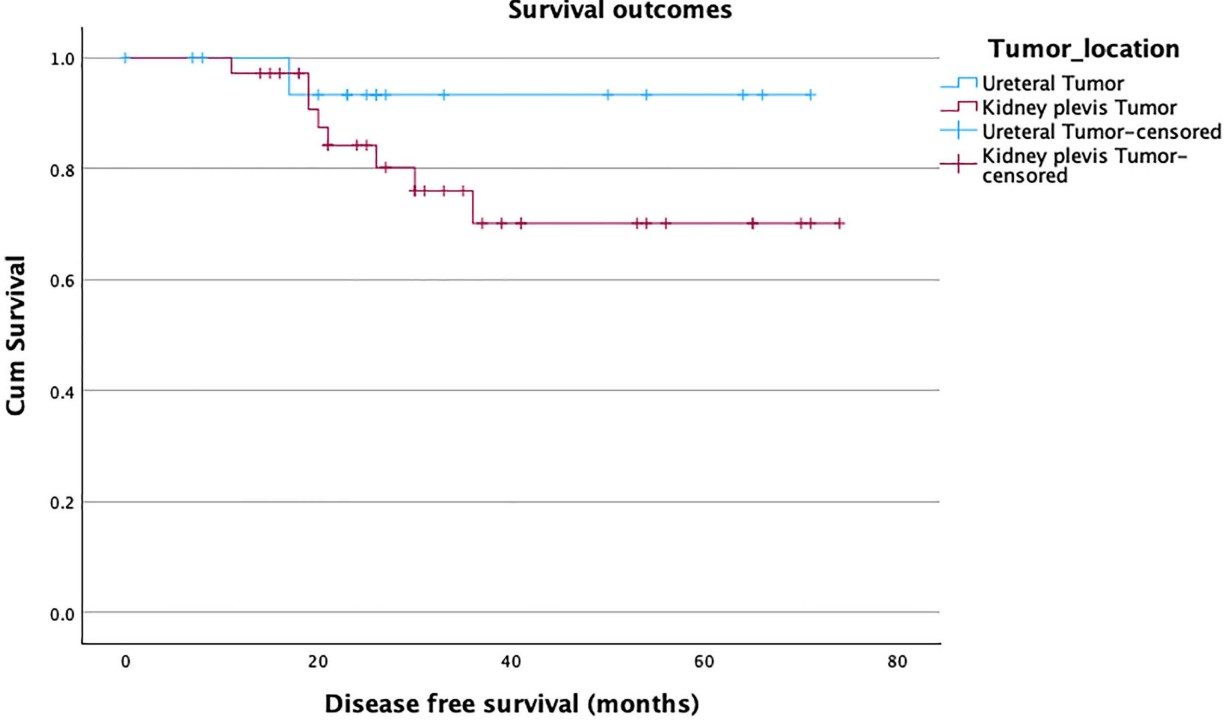

**Fig 2. Kaplan-Meier curve estimating disease-free survival.**

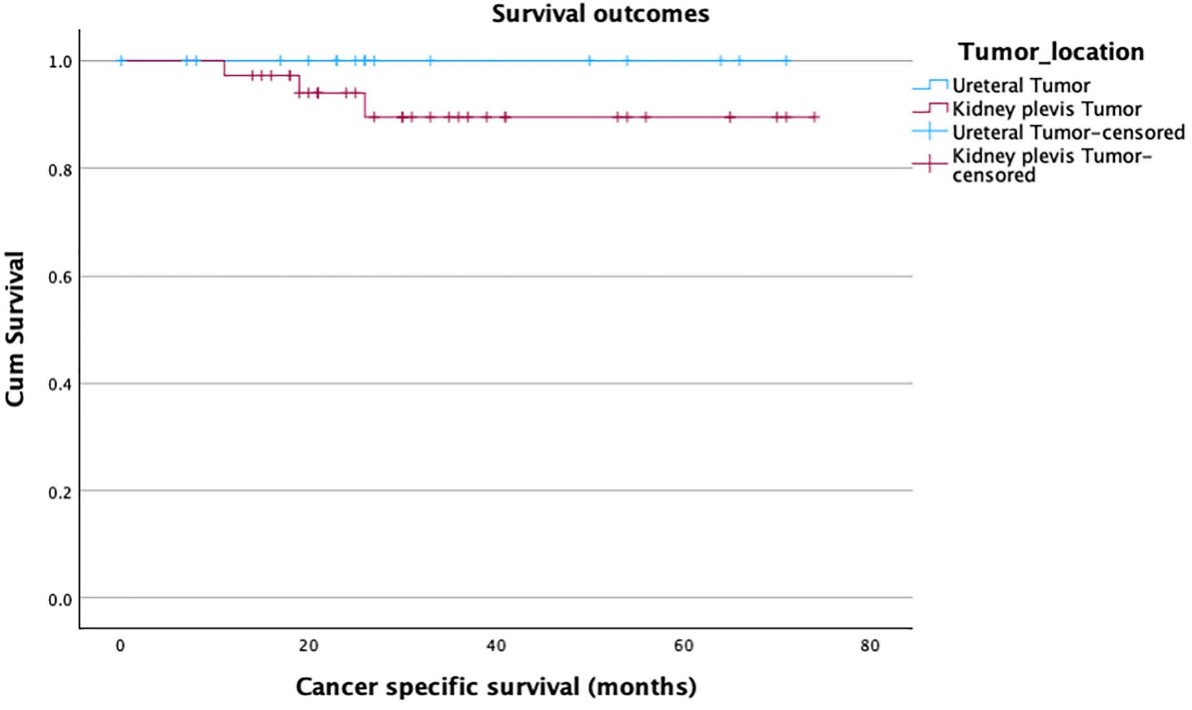

**Fig 3. Kaplan-Meier curve estimating cancer-specific survival.**

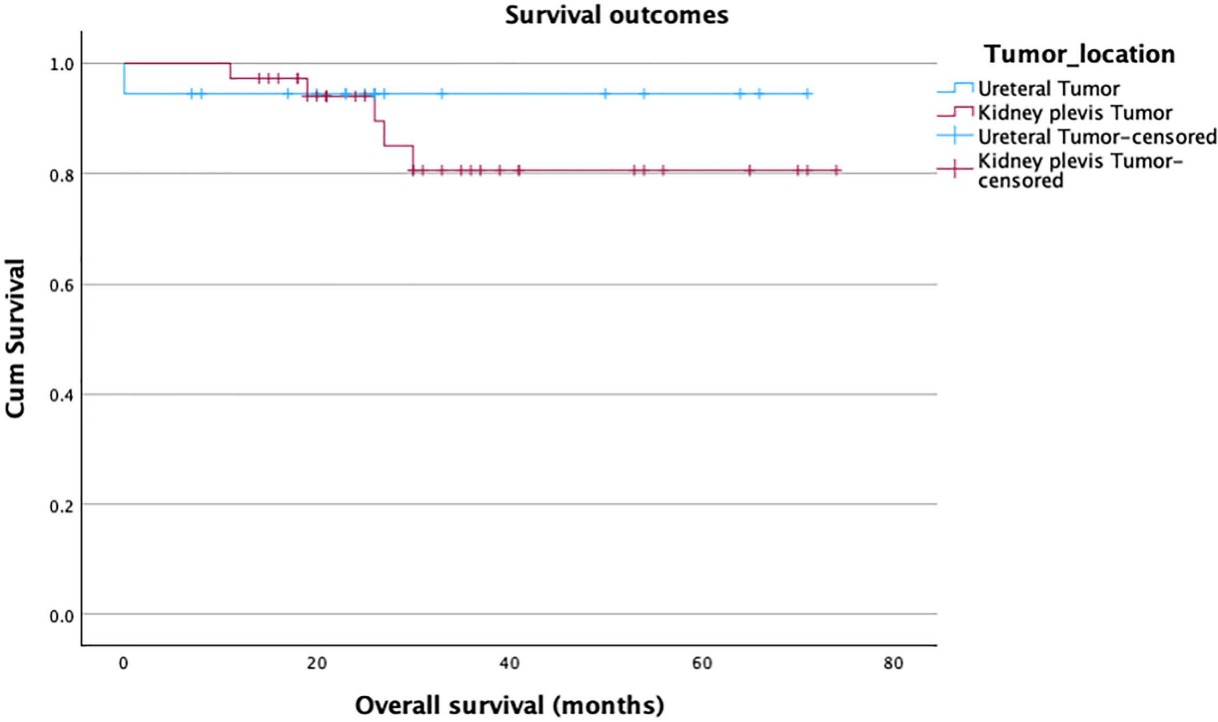

**Fig 4. Kaplan-Meier curve estimating overall survival.**

**Table 3. Complications, readmission, oncological, and survival outcomes.**

| | Total (N = 54) | Ureter (N = 18) | Kidney Pelvis (N = 36) | p-Value |
|---|---|---|---|---|
| Any Complication, n (%) | 11 (20.4%) | 5 (27.8%) | 6 (16.7%) | 0.41 |
| Complication Severity (Clavien-Dindo), n (%) | | | | 0.64 |
| Minor (Grade I-II) | 3 (5.5%) | 2 (11.1%) | 1 (2.7%) | |
| Major (Grade III-V) | 8 (17.0%) | 3 (16.7%) | 5 (13.8%) | |
| 90-Day Readmission, n (%) | 6 (11.1%) | 3 (16.7%) | 3 (8.3%) | 0.42 |
| Oncological Outcomes | | | | |
| Bladder Recurrence Median to recurrance 9 months | 6 (11%) | 1 (5.6%) | 5 (14%) | 0.10 |
| Distant Metastasis Median to metastasis 4 months | 4 (7.4%) | 1 (5.5%) | 3 (8.3%) | 0.72 |
| Survival Outcomes | | | | |
| Disease free Survival | 44 (81%) | 16 (89%) | 28 (77%) | 0.08 |
| Cancer-Specific Survival | 51 (94.4%) | 18 (100%) | 33 (92%) | 0.24 |
| Overall Survival | 48 (89%) | 17 (94.4%) | 31 (86%) | 0.49 |

Categorical data are presented as numbers and percentages. CD: Clavien-Dindo [17].

with 13 receiving adjuvant therapy. The use of adjuvant systemic therapy was evenly distributed between groups, supporting our preliminary conclusion that tumor location did not impact survival.

Furthermore, no significant differences were observed in bladder recurrence, distant metastasis, or overall survival between the groups. Yet ureteral tumor patients showed a higher disease-free survival rate ( 89%) versus (77%) for the renal pelvis group. Nonetheless, the oncological outcomes within our cohort—comprising a bladder recurrence rate of 11%, a distant metastasis rate of 7.4%, and a cancer-specific survival rate of 94.4%—align with the findings documented in currently available literature [8,20]. Except for the disease-free survival, the similar survival outcomes for our study groups, have to be interpreted with caution since our findings may be inflated due to the limited follow-up. And given our small, simple size, our approach carries a high risk of selection bias. It increases the margin of error of our analysis, limiting the ability to generalize the findings to the broader population due to reduced external validity. As a result, our findings may be biased. Another possible explanation for the low recurrence rates observed in our cohort is the absence of positive surgical margins among all patients. This aligns with findings by Colin et al., who reported significantly lower 5-year CSS and metastasis-free survival in patients with positive surgical margins compared to those with negative margins (59.1% and 51.6% vs. 83.3% and 79.3%, respectively) [8,21].

Despite being more obese (median BMI 30.5 kg/m$^2$), patients with ureteral tumors undergoing RANU had a median console time 10 minutes shorter than those with renal pelvis tumors (65 vs. 75 minutes). Both groups received similar lymphadenectomy rates (33% vs. 36%). Interestingly, our overall median operative time of 70 minutes is significantly lower than the durations reported by others [22]. This shorter operative time likely results from avoiding robot redocking, as well as the efficiency and expertise of our surgical team. Chen et al. examined how BMI influences DFS in UTUC patients and found that younger patients with higher BMI were linked to better DFS [23].

In our study, the transfusion rate was 7.4%, consistent with earlier reports (17). Notably, some patients had surgery after extended periods of gross hematuria and received transfusions because of existing comorbidities and to reduce perioperative risks. The median hospital stay was 5 days among both study groups exceeding durations reported in previous studies [24]. This longer stay may reflect local clinical practices, where patients are typically admitted the day before surgery and discharged only after the urinary catheter is removed. Additionally, many older patients often require more time for discharge planning and arranging home care.

Our 90-day readmission rate of 11% corresponds with findings by Liedeberg et al. [25], who reported a rate of 8.2%. However, their reported major complication rate was markedly lower (1/146 patients) compared to ours (8/54 patients). This discrepancy may reflect the impact of the surgical learning curve in our setting, as well as different baseline characteristics in our series in contrast to the Swedish series. In our cohort, the 90-day postoperative mortality rate was 3.7% (2 patients), which is slightly higher than the rates reported in a recent meta-analysis comparing robotic and laparoscopic approaches [19,26]. However, although neither of those two patients was operated on with palliative intent, the two deaths observed in our cohort were likely due to pre-existing vulnerabilities in the patients. In our study, 48% of patients were classified as ASA 3. We also reported additional data, including pre-existing cardiovascular diseases and COPD. However, this classification alone does not fully capture their clinical condition. Consequently, our results should be interpreted with caution and viewed as exploratory rather than definitive.

Apart from the point we discussed, this study has more limitations. In addition to its retrospective design, the relatively small patient group may limit the ability to make meaningful comparisons with other data. However, patients with upper urinary tract cancer who are suitable for surgery are not very common, and it took nearly six years for a high-volume urological department, performing more than 400 robot-assisted surgeries annually, to treat 85 such patients. This underscores the importance of sharing these findings to highlight the challenges in diagnosing and managing these cases. Furthermore, the follow-up period of 26.5 months is too short to allow a thorough analysis of clinical outcomes.

Furthermore, because of the limited number of events, the study has a reduced ability to identify meaningful differences between locations. To improve consistency regarding tumor location among similar patients, we excluded those who underwent segmental ureterectomy or neoadjuvant chemotherapy. We also omitted patients with prior or concurrent cystectomy and those operated on palliatively, as bladder cancer heavily influences survival outcomes.

Additionally, understanding the effects of chemotherapy or immunotherapy is a critical and compelling question; however, it should be examined across different settings and ideally through a randomized clinical trial with an adequate patient sample. This was beyond the scope of this analysis. The final point concerns the Single-Surgeon Series. Although this approach ensures technical consistency and reduces variability, it greatly constrains external validity and generalizability. As a result, the exploratory nature of our findings must be acknowledged. Results achieved by a single expert may not be replicable in lower-volume centers or by surgeons with less experience.

## Conclusion

In our single-center retrospective study of patients undergoing robot-assisted nephroureterectomy for upper tract urothelial carcinoma, considering the small sample size and short follow-up period, no statistically significant difference in survival was observed between ureteral and renal pelvis tumors. While ureteral tumors have previously been associated with poorer prognosis, our data did not confirm this association. The use of a standardized surgical technique may have contributed to the uniformly favorable outcomes observed across both groups. These findings underscore the need for larger, multicenter prospective studies with extended follow-up to better assess the prognostic significance of tumor location in upper tract urothelial carcinoma and to inform individualized treatment strategies.

## Supporting information

**S1 File. RANU-20.12.2025.**
(SAV)

## Author contributions

**Conceptualization:** Mahmoud Farzat.

**Data curation:** Mahmoud Farzat, Ibrahim Altaie.

**Formal analysis:** Mahmoud Farzat.

**Investigation:** Mahmoud Farzat.

**Methodology:** Mahmoud Farzat.

**Project administration:** Mahmoud Farzat.

**Resources:** Mahmoud Farzat.

**Software:** Mahmoud Farzat.

**Supervision:** Mahmoud Farzat, Florian M. Wagenlehner.

**Validation:** Mahmoud Farzat.

**Visualization:** Mahmoud Farzat.

**Writing – original draft:** Mahmoud Farzat.

**Writing – review & editing:** Mahmoud Farzat, Sami-Ramzi Leyh-Bannurah, Mykyta Kachanov, Florian M. Wagenlehner.

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
