## [Decision Letter · Decision Letter 0]

11 Dec 2025

Dear Dr. Farzat,

Thank you for submitting your manuscript to PLOS ONE. After careful consideration, we feel that it has merit but does not fully meet PLOS ONE’s publication criteria as it currently stands. Therefore, we invite you to submit a revised version of the manuscript that addresses the points raised during the review process.

We look forward to receiving your revised manuscript.

Kind regards,

Yudai Ishiyama

Academic Editor

PLOS One

Journal Requirements:

3. Thank you for stating the following in your manuscript:

[Open Access funding enabled and organized by Projekt DEAL.]

[The author(s) received no specific funding for this work.]

6. Please amend the manuscript submission data (via Edit Submission) to include authors Sami-Ramzi Leyh-Bannura, Mykyta Kachanov, and Florian M. Wagenlehner.

7. Please amend your authorship list in your manuscript file to include authors Sami-Ramzi of University Medical Center Hamburg - Eppendorf Leyh-Bannurah, Mykyta of University Medical Center Hamburg - Eppendorf Kachanov, and florian M. m wagenlehner.

8. Thank you for stating the following in the Competing Interests section:

[The authors have declared that no competing interests exist.].

We note that one or more of the authors are employed by a commercial company: UKGM.

Additional Editor Comments:

In addition to the reviewers’ comments, please carefully consider the following points from the Editor and incorporate them into your revision:

1. Scientific soundness and sample size considerations

It is the journal’s policy to consider manuscripts for publication as long as the findings are scientifically sound. I am therefore aiming to make the editorial decision based solely on this criterion. That said, the main finding of the present study—namely, the lack of difference between the two groups—requires careful attention, as underpowered studies often introduce confusion into the scientific literature.

Please review the reviewers’ comments thoroughly, and note that the editorial decision regarding scientific soundness and suitability for publication at this time—especially given the small sample size—will be further considered after your revision.

Additionally, the Introduction requires more context. Many prior studies have evaluated oncological differences between pelvic and ureteral tumors, and this body of evidence should be appropriately summarized.

2. Clarification and standardization of the Methods section

Authors are strongly advised to revise the Methods section using more standardized and conventional descriptions. The current presentation leads to confusion, and clearer methodology will improve the overall readability of the manuscript.

3. Interpretation of non-significant results

As noted by one reviewer, non-significant p-values alone are not valid or clinically meaningful when the sample size is very small. The revised manuscript should avoid overreliance on non-significance and consider alternative approaches to presenting and interpreting the data.

4. Inclusion of figures

Authors are encouraged to provide figures, especially Kaplan–Meier curves. These are essential for adequately presenting survival analyses.

Thank you for your thoughtful revision of the manuscript. I look forward to reviewing your updated submission.

Reviewers' comments:

Reviewer's Responses to Questions

**Comments to the Author**

1. Is the manuscript technically sound, and do the data support the conclusions?

Reviewer #1: Yes

Reviewer #2: Partly

2. Has the statistical analysis been performed appropriately and rigorously?

Reviewer #1: Yes

Reviewer #2: No

3. Have the authors made all data underlying the findings in their manuscript fully available?

Reviewer #1: Yes

Reviewer #2: No

4. Is the manuscript presented in an intelligible fashion and written in standard English?

Reviewer #1: Yes

Reviewer #2: Yes

Reviewer #1: Overall Assessment

This manuscript presents a well-structured retrospective analysis evaluating the impact of tumor location (ureter vs renal pelvis) on perioperative and oncological outcomes following robot-assisted radical nephroureterectomy (RANU). The study population is relatively homogeneous, with all procedures performed by a single experienced surgeon, which strengthens internal validity. The paper is clearly written and provides valuable real-world data regarding surgical outcomes and recurrence patterns.

Importantly, the authors challenge the commonly held view that ureteral tumors carry a worse prognosis, reporting no significant differences between locations in this contemporary RANU cohort. Only minor revisions are required before the manuscript can be considered for publication.

Major Comments

1. The Methods section contains several exclusion criteria (prior cystectomy, segmental ureterectomy, NAC, palliative cases) but the flow of how 85 patients were narrowed to 54 could be more clearly summarized.

Suggestion: Add a short CONSORT-style flow diagram or a clear descriptive paragraph stating the exact numbers excluded for each reason.

2. The manuscript uses the term muscle-invasive (e.g., “15% with muscle-invasive disease”). However, UTUC staging does not commonly use this term (unlike bladder cancer).

Suggestion: Replace with “≥pT2 disease” or specify explicitly that muscle-invasive corresponds to ≥pT2.

3. You state that 33% of patients had “locally advanced disease,” but the staging definition is not provided. Suggestion: Add a definition (e.g., locally advanced = pT3–T4 and/or cN+).

4. It is stated that LAD was performed “in high-risk cases,” but high-risk criteria are not clearly defined.

Suggestion: Specify the criteria (e.g., suspicious nodes on imaging, high-grade tumor, large tumor size).

5. The Results section describes recurrence times, but adding a median time to bladder recurrence—even if the sample is small—would improve clarity.

6. There are several p-values written as “p = 0.9” or “p=0.1” without consistent spacing.

Suggestion: Standardize formatting (e.g., p = 0.90). Also specify whether all p-values are two-sided.

7. CSS is reported as 98.1% and OS as 94.4%, but the follow-up period (median 29 months) is relatively short.

Suggestion: Explicitly highlight in the Results or Discussion that survival estimates may be inflated due to limited follow-up.

8. There are several minor grammatical inconsistencies and spacing issues throughout the manuscript (e.g., “64%%”, “5 lymph nodes”, “10 Adjuvant chemotherapy”).

A light language edit would improve readability.

Reviewer #2: 1. The study is described as a “case-control” study. However, the design is more accurately a retrospective single-center cohort study in which patients are divided into two groups based on tumor location (ureter vs renal pelvis). Please revise the description of the study design throughout the manuscript.

2. The definitions of the survival outcomes and the methods used to assess them are not clearly described. Please explicitly define all primary and secondary endpoints (CSS, OS, RFS).

3. The manuscript states that Kaplan–Meier and Cox regression analyses were used to estimate CSS and OS. However, there is only one cancer-related death and two deaths overall (CSS 98.1%, OS 94.4%). With such a low number of events, Cox regression is statistically unstable and unlikely to yield reliable hazard ratios, and hazard ratios and 95% confidence intervals are not reported. Either present the Cox regression results explicitly (variables included, HRs, 95% CIs) or, if the analysis is not robust, remove Cox regression from the Methods and focus on descriptive Kaplan–Meier curves and univariable log-rank tests. Please clearly state in the Discussion and Limitations that, due to the small number of events, the study has limited power to detect clinically relevant differences between locations.

4. The Methods state that lymphadenectomy was performed in high-risk cases and that all consenting patients received immediate intravesical instillation therapy. In the Results, 26 patients with muscle-invasive or locally advanced disease are reported, of whom 13 received adjuvant therapy (10 chemotherapy, 3 immunotherapy). However, important details are missing: Which agents and doses were used for intravesical instillation? How many patients (overall and by tumor location) actually received it? What were the regimens for adjuvant chemotherapy and immunotherapy? Please provide these details.

5. The median follow-up of 29 months is relatively short for UTUC, particularly for detecting late recurrences and deaths. Nonetheless, the Abstract and Conclusion state that tumor location “does not impact postoperative and oncological outcomes.” Given the small sample and limited follow-up, softening the conclusion is recommended to reflect that no statistically significant differences were observed between ureteral and renal pelvis tumors in this single-center cohort, pending confirmation in larger, multicenter series with longer follow-up.

6. The study reports a statistically significant difference in hospital stay (8 days for the ureter group vs. 6.5 days for the renal pelvis group). The Discussion (Page 8, lines 227-230) attributes this generally to local clinical practices and catheter removal. This finding warrants a more specific explanation of why the ureter group specifically required longer stays, rather than a general comment on discharge planning. Was the protocol for catheter removal different between the groups?

7. There appear to be small inconsistencies between the Abstract and the main text regarding the proportions of muscle-invasive vs locally advanced disease (“14% muscle-invasive” vs “15%” in the Results). Please review and harmonize these percentages.

8. In Table 1, the p-value for gender is 0.5; however, in the first sentence of the Results section, it is described as if the p-value for BMI were 0.5 instead of gender, which seems an error.

9. In Table 1, BMI is significantly higher in the ureteral group than in the renal pelvis group (30.5 vs 27.0 kg/m², p = 0.04). Please comment on the observed baseline difference in BMI and its potential relevance.

10. The Methods state that 85 patients underwent UTUC surgery between July 2019 and July 2025, with data assessed on 01.09.2025. The date of data evaluation is written as “01.09.2025.” To ensure it is correctly understood, please revise it, for example, by writing out the month in English.

11. The term “locally advanced disease,” which seems to refer to T3–T4 tumors, is used in the Discussion, but it is not explicitly defined in the Methods or in the Results. Please clearly define “locally advanced” disease in the Methods.

12. In the Discussion, the statement about Colin et al. and positive surgical margins appears to be cited as reference (16). In contrast, the reference list suggests that Colin et al. correspond to reference 5. Please review all in-text citations and reference numbers for accuracy and consistency.

**Do you want your identity to be public for this peer review?** For information about this choice, including consent withdrawal, please see our Privacy Policy

Reviewer #1: **Yes:** Fumihiko Urabe

Reviewer #2: No

---

## [Author Response · Author response to Decision Letter 1]

22 Dec 2025

Rebuttal letter: Impact of Tumor Location on Oncological and Perioperative Outcomes after Robot-Assisted Radical Nephroureterectomy for Upper Tract Urothelial Carcinoma

Dear Dr. Ishiyama,

Dear Editorial Board of POLS one,

Here is a point-by-point response to your comments and the reviewers' recommendations.

Editor Comments:

In addition to the reviewers’ comments, please carefully consider the following points from the Editor and incorporate them into your revision:

1. Scientific soundness and sample size considerations: It is the journal’s policy to consider manuscripts for publication as long as the findings are scientifically sound. I am therefore aiming to make the editorial decision based solely on this criterion. That said, the main finding of the present study—namely, the lack of difference between the two groups—requires careful attention, as underpowered studies often introduce confusion into the scientific literature.

Please review the reviewers’ comments thoroughly, and note that the editorial decision regarding scientific soundness and suitability for publication at this time—especially given the small sample size—will be further considered after your revision.

Additionally, the Introduction requires more context. Many prior studies have evaluated oncological differences between pelvic and ureteral tumors, and this body of evidence should be appropriately summarized.

We thank you, dear Dr. Ishiyama, for your insightful review and valuable comments.

Regarding sample size and the scientific validity of our study, we clarified in the methods and discussion sections that, in this retrospective context, we used non-probability sampling—specifically, convenience sampling. Given our small, simple size, this approach carries a high risk of selection bias. It increases the margin of error of our analysis, limiting the ability to generalize the findings to the broader population due to reduced external validity. As a result, our findings may be biased. Yet, in this paper, we aimed to reflect our real-world experience with UTUC patients in a high-volume urological department. During a follow-up period of more than 2 years (which is still short for UTUC), we observed no significant differences in survival regarding tumor location in those patients.

First, we revised the introduction and cited an additional four studies to provide more context.

We also conducted a new round of telephone follow-up, repeated the statistical workup, and excluded all patients who died from the calculation of the follow-up period (mean 34 months, median 26,5 months). We rewrote a significant portion of the manuscript. We discovered that two patients had developed distant metastasis, and one of them died due to terminal metastatic disease.

Altogether, in our cohort,

6 patients developed bladder recurrence (median 9 months)

4 patients developed distant metastasis (median 4 months)

2 patients died from surgery-related complications

1 patient died from cancer-unrelated causes

3 patients died from terminal metastatic disease

Changing the survival outcomes into this:

Oncological Outcomes

Bladder Recurrence 6 (11%) 1 (5.6%) 5 (14%) 0.10

Distant Metastasis 4 (7.4 %) 1 (5.5 %) 3 (8.3 %) 0.72

Survival Outcomes

Disease free Survival 44 (81 %) 16 (89 %) 28 (77 %) 0.08

Cancer-Specific Survival 51 (94.4 %) 18 (100%) 33 (92 %) 0.24

Overall Survival 48 (89 %) 17 (94.4 %) 31 (86 %) 0.49

2. Clarification and standardization of the Methods section: Authors are strongly advised to revise the Methods section using more standardized and conventional descriptions. The current presentation leads to confusion, and clearer methodology will improve the overall readability of the manuscript.

As advised, we revised the methodology to improve the manuscript's readability.

We tried to bring more structure in methods by subdividing methods into:

Preoperative diagnostic and Surgical technique

Participants, exclusion criteria, and study design

Variables and Statistical Analysis

3. Interpretation of non-significant results: As noted by one reviewer, non-significant p-values alone are not valid or clinically meaningful when the sample size is very small. The revised manuscript should avoid overreliance on non-significance and consider alternative approaches to presenting and interpreting the data.

We thank you again for the tip. We tried to avoid excessive mention of all p-values in the abstract, results, and discussion. We tried focusing on the relevant outcomes and trends regardless of their p-values.

4. Inclusion of figures: Authors are encouraged to provide figures, especially Kaplan–Meier curves. These are essential for adequately presenting survival analyses.

We are grateful for the valuable recommendation. We added a flowchart for the inclusion criteria. We also included Kaplan-Meier curves for DFS, CCS, and OS. We hope we have improved our manuscripts by presenting the survival analysis.

Reviewers' comments:

Reviewer #1: Overall Assessment

This manuscript presents a well-structured retrospective analysis evaluating the impact of tumor location (ureter vs renal pelvis) on perioperative and oncological outcomes following robot-assisted radical nephroureterectomy (RANU). The study population is relatively homogeneous, with all procedures performed by a single experienced surgeon, which strengthens internal validity. The paper is clearly written and provides valuable real-world data regarding surgical outcomes and recurrence patterns.

Importantly, the authors challenge the commonly held view that ureteral tumors carry a worse prognosis, reporting no significant differences between locations in this contemporary RANU cohort. Only minor revisions are required before the manuscript can be considered for publication.

Major Comments

1. The Methods section contains several exclusion criteria (prior cystectomy, segmental ureterectomy, NAC, palliative cases) but the flow of how 85 patients were narrowed to 54 could be more clearly summarized.

Suggestion: Add a short CONSORT-style flow diagram or a clear descriptive paragraph stating the exact numbers excluded for each reason.

We apologize for the unclear inclusion/exclusion criteria in the methods. We tried to improve the description by adding a consort-style flow diagram to give a better overview of patient inclusion.

2. The manuscript uses the term muscle-invasive (e.g., “15% with muscle-invasive disease”). However, UTUC staging does not commonly use this term (unlike bladder cancer).

Suggestion: Replace with “≥pT2 disease” or specify explicitly that muscle-invasive corresponds to ≥pT2.

3. You state that 33% of patients had “locally advanced disease,” but the staging definition is not provided. Suggestion: Add a definition (e.g., locally advanced = pT3–T4 and/or cN+).

Answer to questions 2 and 3: We removed the term "muscle-invasive disease" and replaced it with ≥pT2 disease, which includes both muscle-invasive and locally advanced disease.

4. It is stated that LAD was performed “in high-risk cases,” but high-risk criteria are not clearly defined.

Suggestion: Specify the criteria (e.g., suspicious nodes on imaging, high-grade tumor, large tumor size).

We defined the high-risk criteria in methods as follows: “Patients with high-grade cytology or biopsy, evidence of local invasion on CT, variant histology, multifocal disease, tumor size ≥2 cm, or hydronephrosis were classified as high-risk UTUC.”

5. The Results section describes recurrence times, but adding a median time to bladder recurrence—even if the sample is small—would improve clarity.

We added the median time to bladder recurrence (9 months) to the abstract and the main text. We also added the median time for the distant metastasis (4 months).

6. There are several p-values written as “p = 0.9” or “p=0.1” without consistent spacing.

Suggestion: Standardize formatting (e.g., p = 0.90). Also specify whether all p-values are two-sided.

We apologize for the inconsistency. We repeated the entire statistical workup, standardized the formatting, and specified in the methods that p-values are two-sided.

7. CSS is reported as 98.1% and OS as 94.4%, but the follow-up period (median 29 months) is relatively short.

Suggestion: Explicitly highlight in the Results or Discussion that survival estimates may be inflated due to limited follow-up.

After we corrected the new CCS and OS rates in our study, we highlighted in the discussion that survival estimates may be inflated due to the limited follow-up period, as follows:

“The oncological outcomes within our cohort—comprising a bladder recurrence rate of 11%, a distant metastasis rate of 7.4%, and a cancer-specific survival rate of 94.4%—align with the findings documented in currently available literature. This finding may be inflated due to the limited follow-up.”

8. There are several minor grammatical inconsistencies and spacing issues throughout the manuscript (e.g., “64%%”, “5 lymph nodes”, “10 Adjuvant chemotherapy”).

A light language edit would improve readability.

We apologize for the inconsistencies. We thoroughly reviewed the manuscript to correct those mistakes.

Reviewer #2: 1. The study is described as a “case-control” study. However, the design is more accurately a retrospective single-center cohort study in which patients are divided into two groups based on tumor location (ureter vs renal pelvis). Please revise the description of the study design throughout the manuscript.

We revised the description of the study throughout the manuscript to “retrospective single-center cohort study.”

2. The definitions of the survival outcomes and the methods used to assess them are not clearly described. Please explicitly define all primary and secondary endpoints (CSS, OS, RFS).

We thank the respected reviewer for his comment. We tried to define our study endpoints in the methods as follows: “The primary endpoint of the study was to assess how UTUC tumor location affects surgical outcomes. The secondary endpoint was to evaluate its influence on disease-free survival and overall survival.”

3. The manuscript states that Kaplan–Meier and Cox regression analyses were used to estimate CSS and OS. However, there is only one cancer-related death and two deaths overall (CSS 98.1%, OS 94.4%). With such a low number of events, Cox regression is statistically unstable and unlikely to yield reliable hazard ratios, and hazard ratios and 95% confidence intervals are not reported. Either present the Cox regression results explicitly (variables included, HRs, 95% CIs) or, if the analysis is not robust, remove Cox regression from the Methods and focus on descriptive Kaplan–Meier curves and univariable log-rank tests. Please clearly state in the Discussion and Limitations that, due to the small number of events, the study has limited power to detect clinically relevant differences between locations.

We thank the respected reviewer for this very important point. As mentioned, due to the small number of events, the Cox regression analysis is not robust and won’t help draw reliable conclusions regarding the impact of UTUC location on survival. We removed the Cox regression from the methods and focused on describing Kaplan-Meier curves with univariable log-rank tests. To improve the interpretation of our results, we included Kaplan-Meier curves for DFS, CCS, and OS in the manuscript. We also stated the following sentence in the limitation (Because of the limited number of events, the study has reduced ability to identify meaningful differences between locations.)

4. The Methods state that lymphadenectomy was performed in high-risk cases and that all consenting patients received immediate intravesical instillation therapy. In the Results, 26 patients with muscle-invasive or locally advanced disease are reported, of whom 13 received adjuvant therapy (10 chemotherapy, 3 immunotherapy). However, important details are missing: Which agents and doses were used for intravesical instillation? How many patients (overall and by tumor location) actually received it? What were the regimens for adjuvant chemotherapy and immunotherapy? Please provide these details.

We apologize for the missed details about the chemotherapy doses and regimes. The intravesical instillation was with mitomycin 40 mg (single dose). We added this information in the methods. The chemotherapy/immunotherapy regimes and doses were described in detail in the methods as follows: “Out of 26 patients with muscle-invasive and locally advanced cancer, 13 received adjuvant therapy. Of them, 10 patients received three cycles of polychemotherapy gemcitabine/cisplatin. When renal function didn’t allow a full dose, patients received a 60% or 80% reduction in cisplatin dose to improve tolerability, and in such cases, the dose was split over two days. Two patients received Nivolumab for 1 year (4 weeks schema), and one patient was planned for adjuvant therapy with nivolumab, yet in the subsequent staging, new metastases were found, and he received Enfortumab Vedotin in combination with Pembrolizumab.”

5. The median follow-up of 29 months is relatively short for UTUC, particularly for detecting late recurrences and deaths. Nonetheless, the Abstract and Conclusion state that tumor location “does not impact postoperative and oncological outcomes.” Given the small sample and limited follow-up, softening the conclusion is recommended to reflect that no statistically significant differences were observed between ureteral and renal pelvis tumors in this single-center cohort, pending confirmation in larger, multicenter series with longer follow-up.

We thank the respected reviewer for his point. We softened the conclusion as follows: “In our single-center retrospective study of patients undergoing robot-assisted nephroureterectomy for upper tract urothelial carcinoma, considering the small sample size and short follow-up period, no statistically significant difference in survival was observed between ureteral and renal pelvis tumors.”

6. The study reports a statistically significant difference in hospital stay (8 days for the ureter group vs. 6.5 days for the renal pelvis group). The Discussion (Page 8, lines 227-230) attributes this generally to local clinical practices and catheter removal. This finding warrants a more specific explanation of why the ureter group specifically required longer stays, rather than a general comment on discharge planning. Was the protocol for catheter removal different between the groups?

To address the points raised by the editor and reviewers, we repeated the entire statistical workup of our study and conducted a new round of follow-up. The median hospital stay across the study groups was 5 days. We corrected those results in both the results and the discussion. And we apologize for the inconsistency.

7. There appear to be small inconsistencies between the Abstract and the main text regarding the proportions of muscle-invasive vs locally advanced disease (“14% muscle-invasive” vs “15%” in the Results). Please review and harmonize these percentages.

We apologize for this inconsistency. We corrected and harmonized the values.

8. In Table 1, the p-value for gender is 0.5; however, in the first sentence of the Results section, it is described as if the p-value for BMI were 0.5 instead of gender, which seems an error.

After repeating the statistical workup of our study, we corrected all the relevant results in the section. Based on the recommendation of the editor, we abstained from mentioning all the non-significant p-values as well. The baseline results now read as follows: “The median age of the patients was 73 years. Patients in the ureteral tumors group were more obese, with a median BMI of 30.5 kg/m², versus a BMI of 27 kg/m² for the renal pelvis group (p = 0.04). 65% of patients were males. Patients presented with multiple comorbidities; however, ASA scores were comparable across groups.”

9. In Table 1, BMI is significantly higher in the ureteral group than in the renal pelvis group (30.5 vs 27.0 kg/m², p = 0.04). Please comment on the observed baseline difference in BMI and its potential relevance.

We integrated the point of higher BMI in discussion as follows: “Despite being

---

## [Decision Letter · Decision Letter 1]

11 Jan 2026

Impact of Tumor Location on Oncological and Perioperative Outcomes after Robot-Assisted Radical Nephroureterectomy for Upper Tract Urothelial Carcinoma

PONE-D-25-54309R1

Dear Dr. Farzat,

We’re pleased to inform you that your manuscript has been judged scientifically suitable for publication and will be formally accepted for publication once it meets all outstanding technical requirements.

Kind regards,

Yudai Ishiyama

Academic Editor

PLOS One

Additional Editor Comments (optional):

All concerns and comments have been adequately addressed. The authors should be commended for their substantial efforts to improve the manuscript.

Reviewers' comments:

Reviewer's Responses to Questions

**Comments to the Author**

Reviewer #2: All comments have been addressed

2. Is the manuscript technically sound, and do the data support the conclusions?

Reviewer #2: (No Response)

3. Has the statistical analysis been performed appropriately and rigorously?

Reviewer #2: (No Response)

4. Have the authors made all data underlying the findings in their manuscript fully available?

Reviewer #2: (No Response)

5. Is the manuscript presented in an intelligible fashion and written in standard English?

Reviewer #2: (No Response)

Reviewer #2: (No Response)

**Do you want your identity to be public for this peer review?** For information about this choice, including consent withdrawal, please see our Privacy Policy

Reviewer #2: No

---

## [Editor Report · Acceptance letter]

PONE-D-25-54309R1

PLOS One

Dear Dr. Farzat,

I'm pleased to inform you that your manuscript has been deemed suitable for publication in PLOS One. Congratulations! Your manuscript is now being handed over to our production team.

Kind regards,

on behalf of

Dr. Yudai Ishiyama

Academic Editor

PLOS One